# MOTO: Offline Pre-training to Online Fine-tuning for Model-based Robot Learning

**Rafael Rafailov**[*][†]     **Kyle Hatch**[*][†]     **Victor Kolev**[†]

**John D. Martin**[‡]     **Mariano Phielipp**[‡]     **Chelsea Finn**[†]

[†]Stanford University   [‡]Intel AI Labs
{rafailov,khatch}@cs.stanford.edu

**Abstract:** We study the problem of offline pre-training and online fine-tuning for reinforcement learning from high-dimensional observations in the context of realistic robot tasks. Recent offline model-free approaches successfully use online fine-tuning to either improve the performance of the agent over the data collection policy or adapt to novel tasks. At the same time, model-based RL algorithms have achieved significant progress in sample efficiency and the complexity of the tasks they can solve, yet remain under-utilized in the fine-tuning setting. In this work, we argue that existing methods for high-dimensional model-based offline RL are not suitable for offline-to-online fine-tuning due to issues with distribution shifts, off-dynamics data, and non-stationary rewards. We propose an on-policy model-based method that can efficiently reuse prior data through model-based value expansion and policy regularization, while preventing model exploitation by controlling epistemic uncertainty. We find that our approach successfully solves tasks from the MetaWorld benchmark, as well as the Franka Kitchen robot manipulation environment completely from images. To our knowledge, MOTO is the first and only method to solve this environment from pixels.[2]

**Keywords:** Model-based reinforcement learning, offline-to-online fine-tuning, high-dimensional observations

## 1 Introduction

Pre-training and fine-tuning as a paradigm has been instrumental to recent advances in machine learning. In the context of reinforcement learning, this takes the form of pre-training a policy with offline learning [1, 2], i.e. when only a static dataset of environment interactions is available; subsequently fine-tuning that policy with a limited amount of online fine-tuning. We study the offline-to-online fine-tuning problem with a focus on high-dimensional pixel observations, as found in real-world applications, such as robotics.

Prior works for offline-to-online fine-tuning often train a policy with model-free offline RL objectives throughout both the offline and online phases [3, 4, 5, 6, 7, 8]. While this approach addresses the challenge of distribution shift in the offline phase, it leads to excessive conservatism in the online phase, since the policy cannot balance offline conservatism with online exploration. Moreover, model-free works are lacking in generalization abilities, and can even under-perform when trained on non task-specific data [9, 10].

---

[*]Equal contribution

[2]Additional details are available on our project website: https://sites.google.com/view/mo2o/

7th Conference on Robot Learning (CoRL 2023), Atlanta, USA.

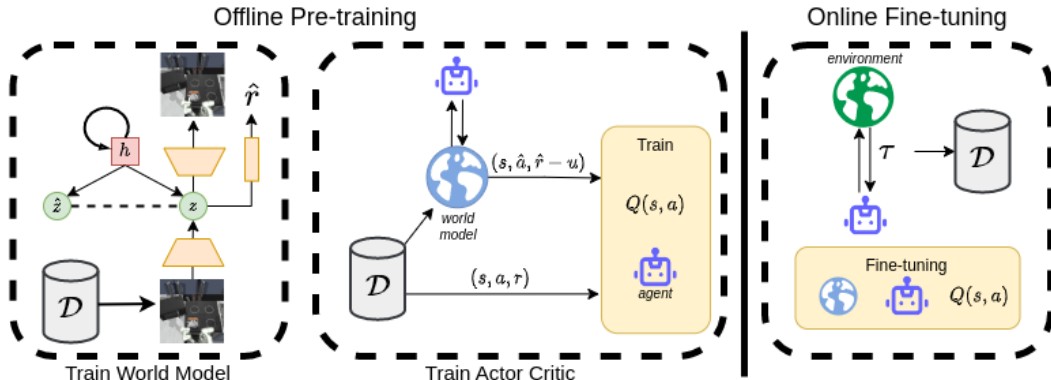

Figure 1: Model-based offline to online fine-tuning. A static dataset of experience is used to train a world model, with which the offline actor-critic agent interacts. The actor-critic agent is trained via both data from the environment and data from the model via model-based value expansion. Model data is penalized via an uncertainty penalty which inhibits model exploitation. Finally, during fine-tuning, the agent interacts with the environment, and collects new trajectories, which are used to jointly fine-tune the world model and actor-critic.

Model-based methods, in which the agent learns a representation and dynamics model, present an interesting alternative, as they have shown generalization ability to new within-distribution tasks [11, 12, 13, 14]; they are sample-efficient and can enable online exploration via model-generated rollouts [15]. Lastly, predictive models can also naturally learn stable representations, which makes them suitable for realistic high-dimensional domains [16, 17, 18, 19]. Despite the good case for model-based learning in the offline-to-online fine-tuning problem, this has been under-explored, with the literature focused mostly on model-free methods.

In this work, we argue that existing algorithms for offline model-based RL are not suitable to the pre-training and fine-tuning or continual learning regimes. In particular, algorithms that use replay buffers of model-generated data, such as [11, 12, 13, 19], create significant distributional shift issues, as the learned model dynamics and reward functions change with additional online interactions. Moreover, models with high-dimensional observations, such as [19, 12], deal with the additional complexity of representation shift of the latent data. These algorithms are also not feasible in large models with high-dimensional representation spaces, which are common in real-world applications [16, 17, 18]. On the other hand, on-policy model-based RL methods such as [20, 21] are amenable to fine-tuning but do not make efficient use of high-quality data in the policy training objective or are not scalable to models with changing representation spaces.

To alleviate these issues, we propose the MOTO (**M**odel-based **O**ffline-**T**o-**O**nline) algorithm. MOTO is a model-based actor-critic algorithm which operates in high-dimensional observation spaces. Crucially, MOTO uses model-based value expansion, which removes the need for large replay buffers, alleviates the distributional shift issue, and allows for the use of large-scale predictive models while still allowing us to use high-quality offline data in the critic learning. To prevent model exploitation, we additionally implement ensemble model-based uncertainty estimation and policy regularization. We evaluate MOTO on 10 tasks from the MetaWorld benchmark [22] and two tasks in the Franka Kitchen domain [23, 24], completely from vision. Our approach outperforms baselines on 9/10 environments in the MetaWorld benchmark and solves both settings in the Franka kitchen. As far as we are aware, is the first method to solve this environment completely from vision. Moreover, by studying the fine-tuning regime, we empirically validate theoretical performance bounds from prior model-based offline RL.

We summarize our contributions as follows: (1) we propose a new model-based actor-critic algorithm for offline pre-training and online fine-tuning; (2) we show the first successful solution to the Franka Kitchen task from images; (3) we empirically verify a proposed theoretical performance gap; (4) to facilitate further research in this area, we will publicly release our environments and datasets.

## 2  Related Work

Our work is at the intersection of offline RL, model-based RL and control from high-dimensional observations (i.e. images). We review related work from these fields below.

**Model-Based Offline RL**    Model-based offline RL algorithms [20, 11, 25, 21, 26, 19, 12] learn a predictive model from the offline dataset and use it for policy training. We would like to design a model-based reinforcement learning algorithm that can efficiently utilize offline datasets, while being easily amenable to continual learning and online fine-tuning. A line of prior works [11, 12, 13] uses MBPO-style optimization [27], which mixes real and model-generated data in a replay buffer used for policy training. [19] generalizes this approach to more realistic domains using variational models and latent ensembles and manages to solve a real robot task involving desk manipulation. However, these methods are not well-suited to the fine-tuning tasks, since the data in the replay buffer is sampled from the model's internal representation space, which suffers from significant distribution shift as the model is fine-tuned. Moreover, the need for replay buffers limits the scalability of these algorithms, as state-of-the-art predictive models in many realistic applications (such as autonomous driving [16, 17, 18]) require very large model and representation sizes. Several algorithms such as MOREL [20] and BREMEN [21] use on-policy training within the learned model without the need for large replay buffers, making them well-suited to continual learning settings, but cannot use potentially high-quality data from the offline dataset to supervise the actor-critic training.

**Variational Dynamics Models**    Variational predictive models have demonstrated success in a variety of challenging applications. One line of research [16, 17, 28, 29, 30] utilizes the model for representation purposes only and uses standard RL, control, or imitation on top of it. Others such as [31, 32, 15, 33] use the latent dynamics model either to learn a policy within the model or deploy shooting-based planning methods. However, most of those prior works focus on the online setting and do not make good use of highly-structured prior data or account for distribution shift. Our method utilizes model-based value expansion, which allows us to take advantage of the efficiency of model-based training, while also using high-quality offline data for critic supervision.

## 3  Preliminaries

In this section we review the modeling framework for our world model and epistemic uncertainty estimates.

**World Model**    To model the high-dimensional observations of the environment, we use a recurrent VAE based on the RSSM model [31, 32]. The model consists of the following components:

$$
\begin{aligned}
\boldsymbol{z}_t &\sim q_\theta(\boldsymbol{z}_t \mid \boldsymbol{h}_t, \boldsymbol{x}_t) && \text{latent representation encoder} \\
\boldsymbol{h}_t &= f_\theta(\boldsymbol{z}_{t-1}, \boldsymbol{h}_{t-1}, \boldsymbol{a}_{t-1}) && \text{deterministic latent state} \\
\hat{\boldsymbol{z}}_t &\sim p_\theta^i(\boldsymbol{z}_t \mid \boldsymbol{h}_t) && \text{stochastic latent state} \\
\hat{\boldsymbol{x}}_t &\sim p_\theta(\boldsymbol{x}_t \mid \boldsymbol{z}_t, \boldsymbol{h}_t) && \text{observation decoder} \\
\hat{\boldsymbol{r}}_t &\sim p_\theta(\boldsymbol{r}_t \mid \boldsymbol{z}_t, \boldsymbol{h}_t) && \text{reward decoder}
\end{aligned}
$$

where $\boldsymbol{x}_t$ are the high-dimensional environment observations, $\boldsymbol{a}_t$ are the actions, $\boldsymbol{r}_t$ are the rewards, $\boldsymbol{h}_t$ are deterministic latent states, and $\boldsymbol{z}_t$ are stochastic latent states. We denote the latent state $\boldsymbol{s}_t = [\boldsymbol{h}_t, \boldsymbol{z}_t]$ as the concatenation of both. All components of the model are trained jointy via the ELBO loss as:

$$
\mathcal{L}_{p_\theta, q_\theta}^{\text{model}} = \underset{\tau \sim \mathcal{D}}{\mathbb{E}} \Big[ \sum_t - \ln p_\theta(\boldsymbol{x}_t \mid \boldsymbol{s}_t) - \ln p_\theta(\boldsymbol{r}_t \mid \boldsymbol{s}_t) +
$$
$$
\mathbb{D}_{KL}[q_\theta(\boldsymbol{s}_t | \boldsymbol{x}_t, \boldsymbol{s}_{t-1}, \boldsymbol{a}_{t-1}) || p_\theta^{i_t}(\boldsymbol{s}_t | \boldsymbol{s}_{t-1}, \boldsymbol{a}_{t-1})] \Big]. \tag{1}
$$

In our experiments we use discrete latent state models, following the DreamerV2 architecture [15]. Notice that we train an ensemble of stochastic latent dynamics models $\{p_\theta^i(\boldsymbol{s}_{t+1} | \boldsymbol{z}_t)\}_{i=1}^M$ following

**Algorithm 1** MOTO: Model-based Offline to Online Fine-tuning

---

**Require:** Offline dataset $D$, initialized policy $\pi_\psi$ and critics $Q_\psi$, initialized prediction and reward model $M_\theta$, policy rollout length $H$, number of offline training steps $N_{\text{offline}}$, number of online episodes to collect $N_{\text{online episodes}}$, number of online gradient updates per episode $G$.

1: *// offline pre-training*
2: **for** $i = 1, 2, 3, \cdots, N_{\text{offline}}$ **do**
3:      Sample a batch of trajectories $B \sim \mathcal{D}$.
4:      Update $M_\theta$ on $B$ according to Eq. 1.
5:      Generate $H$-step latent policy rollouts with penalized rewards (Eq. 3).
6:      Update $\pi_\psi$ according to Eq. 10
7:      update $Q_\psi$ according to Eq. 9 on the real and model data.
8: **end for**
9: *// online fine-tuning*
10: **for** $i = 1, 2, 3, \cdots, N_{\text{online episodes}}$ **do**
11:      Rollout the policy $\pi_\theta$ in the environment for an episode to collect a new trajectory $\tau$
12:      $\mathcal{D} = \mathcal{D} \cup \tau$
13:      **for** step $= 1, 2, 3, \cdots, G$ **do**
14:          Sample a batch of trajectories $B \sim \mathcal{D}$.
15:          Update $M_\theta$ on $B$ according to Eq. 1.
16:          Generate $H$-step latent policy rollouts with penalized rewards (Eq. 3).
17:          Update $\pi_\psi$ according to Eq. 10
18:          update $Q_\psi$ according to Eq. 9 on the real and model data.
19:      **end for**
20: **end for**

---

[19] by randomly selecting one model $p_\theta^{i_t}$ to optimize at each time step of the trajectory in the model in Eq. 1. This makes the ensemble training no more computationally expensive than single model optimization.

**Offline Model-Based RL From High-Dimensional Observations**    To mitigate issues with model exploitation, similar to [19, 11], we train a latent dynamics model ensemble $\{p_\theta^i(s_{t+1}|z_t)\}_{i=1}^M$, and implement model conservatism by penalizing rewards via dynamics model disagreement, which acts as a proxy for epistemic uncertainty. We use the penalty:

$$u_\theta(s_t, a_t) = \text{std}(\{l_{\theta^i}(z_{t+1})\}_{i=1}^M), \tag{2}$$

where $l_\theta^i(z_{t+1})$ is the logit outputs of the discrete distribution $p_\theta^i(\cdot|z_{t+1})$. Hence, the final reward function is

$$\hat{r}_\theta(s_t, a_t, s_{t+1}) = r_\theta(s_{t+1}) - \alpha u_\theta(s_t, a_t) \tag{3}$$

where $\alpha$ is a trade-off parameter between reward maximization and conservatism.

## 4   Model-based Offline to Online Fine-tuning (MOTO)

Our model training architecture and objective follow prior works [19, 34], but we significantly change the actor-critic algorithm optimization towards the goal of efficient online fine-tuning from offline data. We build the MOTO policy optimization procedure on three main design choices: 1) model-based value expansion, 2) uncertainty-aware predictive modelling, and 3) behaviour-regularized policy optimization. The full algorithm training outline is presented in Algorithm 1.

**Variational Model-Based Value Expansion**    We would like to train a policy via model-based training and without latent replay buffers, while making use of both the high-quality offline data and the world model as sources of supervision for the actor and critic. To this end, we adapt ideas from the model-based value expansion literature [35, 36, 37, 38]. We consider sequences of data of the form $\tau = (x_{1:T}, a_{1:T}, r_{1:T})$. At each agent training step, we infer latent states $s_{1:T}^0 \sim q_\theta(s_{1:T}|x_{1:T}, a_{1:T})$. We then use the true data as starting points for model-generated rollouts, as:

$$\hat{a}_j^t \sim \pi_\psi(a|\hat{s}_j^{t-1}), \quad \hat{s}_j^{t+1} \sim p_\theta(s|\hat{a}_j^t, \hat{s}_j^t), \quad \hat{r}_j^t \sim p_\theta(r|\hat{s}_j^t), \tag{4}$$

where the rewards are computed according to Eq. 3. Following standard off-policy learning algorithms, we use critics $\{Q_{\psi^1}, Q_{\psi^2}\}$ and and target networks $\{\bar{Q}_{\psi^1}, \bar{Q}_{\psi^2}\}$. We can then use our model to estimate Monte-Carlo based policy returns:

$$V_0^{\pi_\psi}(\hat{\boldsymbol{s}}_j^t) = \min\{Q_{\psi^1}(\hat{\boldsymbol{s}}_j^t, \hat{\boldsymbol{a}}_j^t), Q_{\psi^2}(\hat{\boldsymbol{s}}_j^t, \hat{\boldsymbol{a}}_j^t)\}, \quad V_K^{\pi_\psi}(\hat{\boldsymbol{s}}_j^t) = \sum_{k=1}^{K} \gamma^{k-1} \hat{\boldsymbol{r}}_j^{k+t} + \gamma^K V_0^{\pi_\psi}(\hat{\boldsymbol{s}}_j^{t+K})$$

And compute the GAE$(\gamma, \lambda)$ estimate:

$$V^{\pi_\psi}(\hat{\boldsymbol{s}}_j^t) = (1 - \lambda) \sum_{k=1}^{H-t-1} \lambda^{k-1} V_k^{\pi_\psi}(\hat{\boldsymbol{s}}_j^t) + \lambda^{H-t-1} V_{H-t}^{\pi_\psi}(\hat{\boldsymbol{s}}_j^t) \tag{5}$$

We denote $\widehat{V}^{\pi_\psi}(\boldsymbol{s}) := \lambda V^{\pi_\psi}(\boldsymbol{s}) + (1 - \lambda) V_0^{\pi_\psi}$, and optimize the actor objective as:

$$\mathcal{L}_{\pi_\psi}^{\text{model}} = -\frac{1}{HT} \mathbb{E}_{\tau \sim \mathcal{D}} \mathbb{E}_{\pi_\psi, p_\theta} \left[ \sum_{t=0, j=1}^{H-1, T} \widehat{V}^{\pi_\psi}(\hat{\boldsymbol{s}}_j^t) \right] \tag{6}$$

This objective essentially estimates the actor return by mixing Monte-Carlo based estimates at various horizons. Notice that is a fully differentiable function of the policy parameters, by backpropagating through the Q-functions and dynamics model. Also notice that for $H = 0$ this is just the standard actor-critic policy update.

We can similarly use MC return estimates to train the critics. We recompute the critic target values $\bar{V}^k(\hat{\boldsymbol{s}}_j^t)$ for all states similarly to Eq. 5 using the target networks $\{\bar{Q}_{\psi^1}, \bar{Q}_{\psi^2}\}$. The critics are trained on both the model-generated and real data with the sum of two losses:

$$\mathcal{L}_{Q_{\psi^i}}^{\text{model}} = \frac{1}{HT} \mathbb{E}_{\tau \sim \mathcal{D}} \mathbb{E}_{\pi_\psi, p_\theta} \left[ \sum_{t=0, j=1}^{H-1, T} (\bar{V}^{\pi_\psi}(\hat{\boldsymbol{s}}_j^t) - Q_{\psi^i}(\hat{\boldsymbol{s}}_j^t, \hat{\boldsymbol{a}}_j^t))^2 \right] \tag{7}$$

$$\mathcal{L}_{Q_{\psi^i}}^{\text{data}} = \frac{1}{T-1} \mathbb{E}_{\tau \sim \mathcal{D}} \mathbb{E}_{\pi_\psi} \left[ \sum_{j=1}^{T-1} \left( \boldsymbol{r}_{j+1}^0 + \gamma \widehat{V}^{\pi_\psi}(\boldsymbol{s}_{j+1}^0) - Q_{\psi^i}(\boldsymbol{s}_j^0, \boldsymbol{a}_j^0) \right)^2 \right] \tag{8}$$

(notice that the second equation does not involves dynamics model samples) and the final critic loss is the sum of the two:

$$\mathcal{L}_{Q_{\psi^i}}^{\text{final}} = \mathcal{L}_{Q_{\psi^i}}^{\text{model}} + \mathcal{L}_{Q_{\psi^i}}^{\text{data}} \tag{9}$$

Training the critic networks on the available offline data serves as a strong supervision when the dataset already contains rollouts with high returns.

**Uncertainty-aware Predictive Modelling** In order to prevent model exploitation in the offline training, we use model-based uncertainty estimates via ensemble statistics, similar to [39, 40, 41, 42, 43, 44, 45, 46]. Note that the loss $\mathcal{L}_{Q_{\psi^i}}^{\text{data}}$ is computed on transitions sampled from the dataset trajectories through the inference model $q_\theta$ and have ground-truth environment rewards. In contrast, the critic loss $\mathcal{L}_{Q_{\psi^i}}^{\text{model}}$ is computed on synthetic states sampled from the model using only uncertainty-penalized rewards (Eq 2). This explicitly builds conservatism into the critic values by biasing them towards dataset states and actions. We also considered alternative conservative critic optimization [4, 12]. However, these approaches are incompatible with multi-step returns and require the use of a latent replay buffer which is undesirable. Following prior works [11, 20, 19] we provide theoretical verification for our modelling choices. In addition, by studying the online fine-tuning regime, for the first time, we are able to provide empirical verification for prior offline MBRL performance bounds. Since the current work does not focus on theoretical contributions, we defer these results to Appendix B.

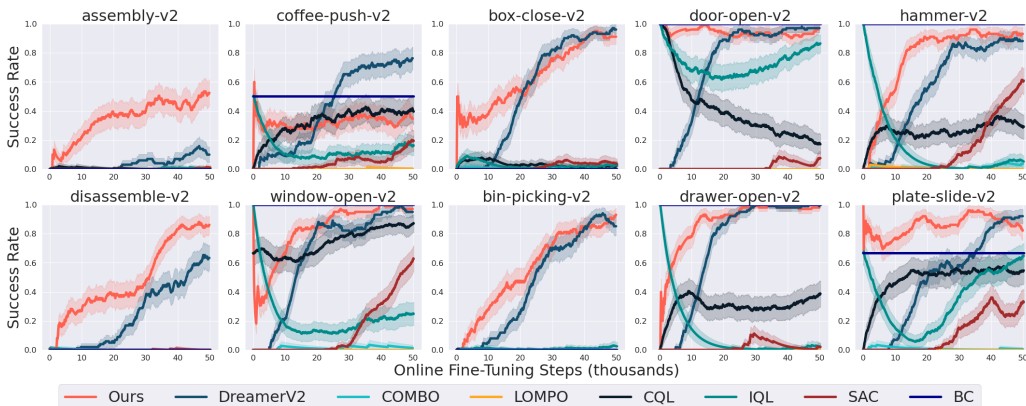

Figure 2: The success rates across the 10 MetaWorld tasks. MOTO matches or outperforms other methods on 9 out of the 10 tasks, demonstrating MOTO's ability to successfully pre-train offline and fine-tuning online on a variety of manipulation tasks using limited offline data. DreamerV2 is the only other method to achieve competitive results on the MetaWorld tasks. The model free baselines achieve low to moderate performance across all tasks.

**Behaviour Prior Policy Regularization**   Realistic robot learning datasets often consist of narrow data like planner based rollouts or human demonstrations. As such, at the initial stages of training dynamics models can be quite inaccurate and the agent can benefit from stronger data regularization for the policy [26, 13, 25, 21]. To avoid additional complexity of modelling the behaviour distribution we follow an approach similar to [47] which deploy a regularization term of the form

$$\mathcal{L}^{\text{reg}}_{\pi_\psi} = - \mathop{\mathbb{E}}_{\tau \sim \mathcal{D}} \left[ \sum_{t=1}^{T} \log \pi_\psi(\boldsymbol{a}_t \mid \boldsymbol{s}_t) f \left( \underbrace{\gamma^H V^{\pi_\psi}(\boldsymbol{s}_{t+H}) + \sum_{j=1}^{H} \gamma^j \boldsymbol{r}_{t+j} - V^{\pi_\psi}(\boldsymbol{s}_t)}_{\text{Advantage over trajectory snippet } \boldsymbol{s}_t \,:\, \boldsymbol{s}_{t+H}} \right) \right]$$

for some function $f$. The authors suggest a simple threshold function works well (i.e adding a behaviour cloning term to snippets with positive advantage), [13] can also be viewed as an instance of this approach using exponential weighting. In this work we focus on realistic robot manipulation tasks with sparse rewards, and just threshold trajectories based on whether they achieve the goals in the environment. We then optimize the joint actor loss:

$$\mathcal{L}_{\pi_\psi} = \mathcal{L}^{\text{model}}_{\pi_\psi} + \beta \mathcal{L}^{\text{reg}}_{\pi_\psi} \tag{10}$$

where $\beta$ is a hyper-parameter that trades-off between model-based optimization and data regularization.

## 5   Experiments and Results

We aim to answer the following questions: (1) Can MOTO pre-train offline and successfully fine-tune online? (2) What is the impact of different model components? (3) Does MOTO exhibit good generalization and sample efficiency?

**Experiment Setup**   We evaluate our method on two challenging dexterous manipulation domains, MetaWorld [22] and the Standard Franka Kitchen environment [24, 23] used in the D5RL benchmark [48].

MetaWorld contains a variety of simulated manipulation tasks in a shared, table-top environment to be solved with a Sawyer robotic arm. We use ten of these tasks for our experiments (see Figure 7). We modify these environments to use 64x64 RGB image observations, without any robot proprioception and use sparse rewards based on task completion. For each environment we collected a small dataset of 9-10 demonstration episodes using a scripted policy.

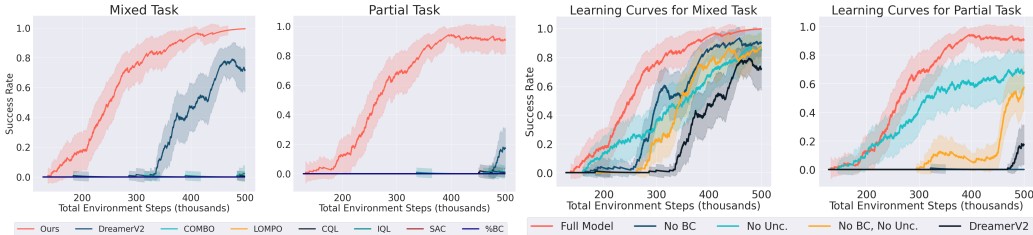

Figure 3: (**Left**) Success rate of completing the "mixed" and "partial" tasks in Franka Kitchen. MOTO outperforms all methods on both tasks, and is the only method to achieve meaningful progress on the "partial" task, indicating MOTO's capacity for combinatorial generalization. (**Right**) We carry out ablations on the MOTO design: no uncertainty penalties "No Unc.", no behavioral cloning regularization "No. BC", and removing both "No BC., No Unc."; removing model-based value expansion as well gives us DreamerV2. We observe that the gains from each component are additive, and only the full model achieves full performance. Lastly, since all ablations share the same architecture, this shows that the performance improvement is not due to a stronger architecture, but rather the actor critic training.

We also evaluate MOTO on the Standard Franka Kitchen environment from the D5RL benchmark [48], which is a challenging long-range control problem that requires using a simulated 9-DOF Franka Emika Robot to manipulate multiple different objects in a simulated kitchen area. For our experiments, we only use the central camera image without the wrist camera view or robot propriocpetion. Since the "partial" task does not contain successful trajectories for all four target objects, we only regularize policy training with respect to the first three objects.

More details about the environments and datasets can be found in Appendix C.

**Baselines**   We compare our method to prior vision-based offline model-based RL algorithms LOMPO [19] and COMBO [12] as well as DreamerV2 [15], a state-of-the art online model-based learning algorithm. We also compare our approach against CQL, [4] a successful model-free offline RL algorithm, IQL, [3] a state-of-the art model-free regression-based fine-tuning algorithm, SAC [49], and behaviour cloning. All methods are pre-trained offline for 10 thousand gradient steps and fine-tuned with online interactions for a total of 500 thousand environment steps.

**MetaWorld Results**   Results for the MetaWorld tasks are in Fig. 2. MOTO outperforms other methods on 9 out of 10 tasks in that domain. This shows MOTO's ability to successfully pre-train offline and fine-tune online on a variety of manipulation tasks using limited offline data. DreamerV2 is the only other method to achieve competitive results on the MetaWorld tasks. The model free baselines achieve low to moderate performance across all tasks. Perhaps somewhat surprisingly, COMBO and LOMPO achieve very low success rates on most of the tasks. A speculative explanation for this is that the MetaWorld environments have a significant degree of randomization between them at each new episode, causing the learned image representations to change frequently. Since COMBO and LOMPO are off-policy methods that maintain replay buffers of the latent state representations, if these representations change frequently this would lead to poor performance.

**Franka Kitchen Results**   As seen in Fig. 3, our method successfully solves both the "mixed" and "partial" tasks of Franka Kitchen with 100% and 90.5% final success rates respectively. DreamerV2 is the only other method to obtain non-trivial success rates. Noteworthy is MOTO's success on the "partial" task, which demonstrates that the world model is capable of combinatoric generalization.

While model-free methods make some progress, ultimately they stagnate and cannot successfully complete all four objects on either task. This is likely due to the partial observability of the environment, since the robot can occlude manipulated objects and also requires joint state estimation directly from images. In contrast, variational models serve as Bayesian filters and naturally build state estimations of the environment in the latent space. Model-based methods LOMPO and COMBO achieve very limited progress, due to the non-stationarity issues described in the beginning of Section 4. The DreamerV2 algorithm learns more slowly and only reaches final success rates of 77.5%

and 13.5% versus 100% and 90.5% for our method on the "mixed" and "partial" task. As far as we know, our model is the first method to solve the Franka Kitchen environment from images.

**Ablation Studies**   In this section we evaluate the contribution of each model component to final performance. Results are presented in Fig. 5 (right). We also include the standard DreamerV2 algorithm for direct model-based comparison. While all ablations make significant progress on the "mixed" task, only the full model manages to solve it entirely. The full model outperforms the others on the "partial" task by a very significant margin. We also note that without any behavioral cloning (BC) data regularization, both the "No BC., No Unc." and DreamerV2 methods learn unsafe behaviours, such as hitting the kettle into the goal position, or smashing the light switch with the robot head, instead of using its gripper to grasp and place the objects. These policies would be unsafe for both the hardware and environment in a real setting. Such behaviours are not present in any of the regularized methods (videos are available on the project website).

# 6   Discussion

**Future Work**   The MOTO algorithm design does not require large replay buffers of intermediate representations, while still allowing the use of high-quality data to supervise the critic learning and bootstrap the policy optimization. We believe that these qualities make MOTO very suitable for realistic applications, which require large scale models [16, 17, 18]. We plan to evaluate MOTO on large-scale realistic domains, such as CARLA [50] in future work.

MOTO is also well-suited to the model-based imitation learning setting [51, 14, 52, 53], which has recently been successfully applied to real world scenarios as well [54, 55]. By using on-policy roll-outs, MOTO can maintain the stability and theoretical guarantees of adversarial imitation learning [56, 57, 19], while still using the high-quality expert data to both provide supervision to the critic, as well as to regularize the policy.

**Limitations**   MOTO builds a level of pessimism through penalizing state-action epistemic model uncertainty. Excessive pessimism can prevent the model from exploring or generalizing outside of the available data distribution. This can hurt performance if the offline dataset consists of lower-quality or incomplete data. MOTO also uses policy regularization, which is based on task success. It may be difficult to adapt this approach to tasks that involve more complex or non-sparse rewards. Finally, a key component of MOTO is controlling model-based epistemic uncertainty. We train an ensemble of latent transition models and use their disagreement as a reward penalty. The models we consider use MLPs for the latent dynamics, however it is not clear if this scheme can transfer to more complex architectures, such as Transformers, which are now more widely used within the predictive modeling context.

# 7   Conclusion

We present MOTO, a model-based reinforcement learning algorithm specifically designed for the offline pre-training and down-stream fine-tuning regime. MOTO learns a variational model directly from pixels, and trains an actor-critic agent within the learned latent dynamics model using model-based value expansion, epistemic uncertainty corrections, and policy regularization, all of which we show have an impact in improving performance, and deriving safe and robust policies. MOTO outperforms baselines in terms of sample efficiency and final performance on 9/10 MetaWorld tasks and far as we are aware is the first and only method to solve the Franka Kitchen benchmark from images. Furthermore, by studying the offline pre-training and fine-tuning regime, we empirically verify long-standing theoretical results on the offline model-based RL problem. Finally, the structure of the algorithm makes it suitable for use in very large scale dynamics models, such as the ones used in autonomous driving, which prior work was not able to utilize, as well as a backbone for model-based imitation, multi-task and transfer learning. We plan to explore these directions in follow-up works.

**Acknowledgments**

We would like to express gratitude to the reviewers, whose feedback helped improve this paper. Chelsea Finn is a CIFAR Fellow in the Learning in Machines and Brains program. This work was supported by Intel AI Labs and ONR grant N00014-21-1-2685.

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

# A  Additional Experiments

## A.1  Model-Based Generalization

The "partial" task also provides a good test bed for
an algorithm's generalization capabilities, since the
offline dataset does not contain full solutions for
it. This is a different problem than the standard
dynamic programming ("stitching") issue of data-
centric reinforcement learning since the dataset does
not contain a sequence of state-action pairs that lead
from the initial state to the goal state. Instead, to
solve this task, a learning agent must understand
the compositional nature of the scene and do com-
binatorial generalization over the objects. In this
section we seek to answer whether 1) the learned
model can do combinatorial generalization of within
distribution tasks and 2) whether policy optimiza-
tion can take advantage of the model's capabilities.
We evaluate the agent at the end of the offline pre-
training phase. To answer the first question, we con-
sider episodes that successfully complete the "par-
tial" task from the trained agent. We condition our
model on the frames that solves the first three tasks

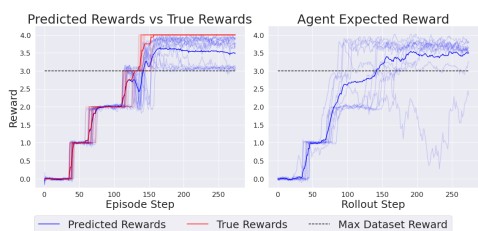

Figure 4: We evaluate the model's gener-
alization capabilities at the end of the of-
fline pre-training phase. The model correctly
predicts rewards of up to 4 on successful
episodes in the "partial" task, even though
the maximum dataset reward is 3. (left).
When doing rollouts in the learned model,
the policy solves all four objects in the "par-
tial" task and reaches rewards of up to 4
(right).

(which are covered in the offline dataset) and rollout the expert actions to predict the following
frames. Results are shown in Fig. 1. The model successfully predicts a combination of the mi-
crowave, kettle, bottom burner and light switch in the correct configuration, despite never encoun-
tering these four objects together in the offline dataset. Moreover, we evaluate the model-predicted
rewards on these expert trajectories, plotted in Fig. 4 (left). We see that the model predicts rewards
of up to 4 with an average reward of 3.63, despite only being trained on trajectories with maximum
reward of 3. This results show that the learned model is capable of compositional generalization. To
evaluate whether the learned policy can take advantage of the model generalization capabilities, we
rollout the trained agent under the model and evaluate the predicted rewards, results are shown in
Fig. 4 (right). The agent achieves an average final reward of 3.52 under the learned model and solves
all four tasks. This suggest that the model-based RL agent is able to do combinatorial generalization,
but the offline dataset is not enough to adequately learn the environment dynamics.

## A.2  Constrained Offline Data Ablation

We aim to test the performance of MOTO under a data-
constrained regime, evaluating whether learning slows
down with less offline data. To do so, we randomply
sampled 100 and 250 episodes from the Franka Kitchen
dataset and used them for offline training, evaluating on
the "Mixed" task. The results are presented in fig. 5. We
observe that learning is not slowed down by reduction in
offline data, at least to the extent that we tested. This
shows that MOTO is robust to a constrained set of offline
data, and can operate at the same performance level even
with 5 times less offline episodes. Notably, we hypothe-
sise there is a minimum threshold for diversity of offline
learning, yet we see no performance degredation even at
100 episodes. It is important to point out that episodes
were randomly sampled without regards to the reward at-

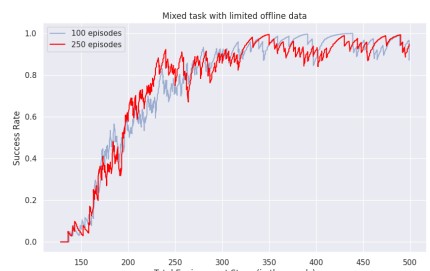

Figure 5: Training curves for data ab-
lation experiments. We see no degra-
dation in performance when using only
100 and 250 pre-training episodes.

tained in each episode, i.e. the 100 episodes are not of proportionally higher quality.

# B Theoretical Results and Empirical Validation

**Theoretical Results for Uncertainty-Aware Model-based Training** Given our choice of variational parametrization and model uncertainty estimation we can directly adapt certain theoretical guarantees from prior model-based RL literature [11, 20, 19]. We consider the following result in particular: let $T_\theta(s'|s, a)$ and $T(s'|s, a)$ be the learned and true latent dynamics models respectively. We define the discounted state-action distribution

$$\rho^\pi_{\mathcal{T}, \mu_0}(s, a) \propto \sum_{t=0}^\infty \gamma^t \mathbb{P}^\pi_{\mathcal{T}, \mu_0}(s_t = s) \pi(a|s)$$

in the standard way. The function $u(s, a)$ is an admissible error estimator if

$$d_\mathcal{F}[T(s'|s, a)||T_\theta(s'|s, a)] \leqslant u(s, a).$$

For any policy $\pi$ we can then define

Figure 6: Empirical evaluation of Theorem B.1. We plot the performance gap versus the the empirical estimates of (normalized) expected model uncertainty using Eq. 12.

$$\epsilon_u(\pi) = \mathbb{E}_{(s,a) \sim \rho^\pi_{T_\theta, \mu_0}} [u(s, a)].$$

The following Theorem then holds:

**Theorem B.1.** *(Informal) Let $\hat{\pi}^*(s)$ be the optimal policy under the learned model $T_\theta(s'|s, a)$ with an uncertainty-penalized reward and $\pi^*$ the optimal policy in the ground-truth MDP. Under certain mild assumptions, then the following inequality holds:*

$$2\alpha\epsilon_u(\pi^*) \geqslant \mathbb{E}_{\pi^*, T}\Big[ \sum_{t=0}^\infty r_t \Big] - \mathbb{E}_{\hat{\pi}^*, T}\Big[ \sum_{t=0}^\infty r_t \Big] \tag{11}$$

*Proof.* Consult [11]. □

**Empirical verification** From the Theorem, we can deduce that the policy under-performance is upper bounded by the discounted model-based uncertainty over the state-action distribution induced by the expert policy under the learned model. In practice we do not have access to an oracle estimator $u(s, a)$ and we use the ensemble disagreement from Eq. 2. While these results are not new, empirical verification is difficult in the fully offline case, since we have a static dataset, and all values are point estimates. However, in the online fine-tuning case, we have a continuum of datasets and we can empirically verify the claims of Theorem B.1.

We periodically evaluate $\epsilon_u(\pi^*)$ and the expected model uncertainty induced under the expert state-action distribution in the learned model. At each epoch $E$, we cannot generate model rollouts from the expert, since that would require training an expert policy under the current inference model $q_{\theta_E}$. However, we can sample expert episodes from the trained expert and the environment. Given an expert trajectory $\tau^{\text{exp}} = x_{1:T}, a_{1:T}$ we sample latent belief states from the first $T - H$ steps to obtain $s_{1:(T-h)} \sim q_{\theta_E}(\cdot|x_{1:T-H}, a_{1:T-H})$. From each state $s_j$ we then rollout the expert actions $a_{j:j+H}$ using the current iteration of the dynamics model $T_{\theta_E}$ and obtain states $\{(\hat{s}_j^t, a_j^t)\}_{j=1,t=0}^{T-H,H}$ as in Section 4 (here $a_j^t = a_{j+t}$ from the expert dataset. We can then obtain the empirical estimate of

$$\epsilon_u(\pi^*) \approx \mathbb{E}_{q_{\theta_E}(s_j^0|\tau^{\text{exp}}), T_{\theta_E}}\Big[ \frac{1}{H(T-H)} \sum u_\theta(\hat{s}_j^t, a_j^t) \Big] \tag{12}$$

Empirical results evaluated on the "partial" task are shown in Fig. 6. We see that the performance gap is strongly bounded (up to a choice of the penalty scale) by the estimate from Eq. 12, which verifies the claim of Theorem B.1.

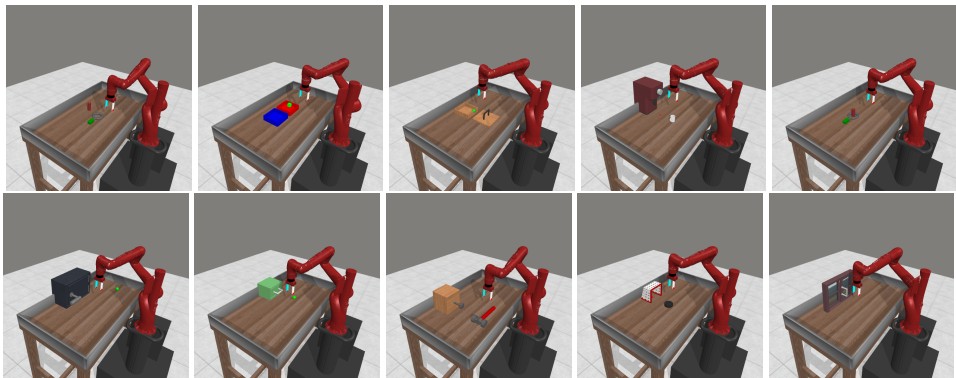

Figure 7: Visualization of the 10 different MetaWorld environments used in our experiments. Top row from left to right: `assembly-v2`, `bin-picking-v2`, `box-close-v2`, `coffee-push-v2`, `disassemble-v2`. Bottom row from left to right: `door-open-v2`, `drawer-open-v2`, `hammer-v2`, `plate-slide-v2`, `window-open-v2`.

## C  Experimental Details

### C.1  Environments

The Franka Kitchen environment from [23] (RPL) is a challenging long-range control problem, which involves a simulated 9-DOF Franka Emika Robot in a kitchen setting. The robot uses joint-space control and the observation is a single 64x64 RGB image; we do not assume access to object states or robot proprioception. The goal of the agent is to manipulate a set of 4 pre-defined objects and receives a reward of 1.0 for each object in right configuration at each time step. This is a very challenging environment due to 1) high-dimensional observation spaces; 2) partial observability with non-trivial object and robot state estimation; 3) need for very-fine-grained control in order to operate the small elements of the environment, such as turning knobs and flipping the light-switch; 4) the long-range nature of the tasks; 5) the use of sparse rewards, which provide limited intermediate supervision to the policy, and finally 6) the use of high-dimensional control which requires learning forward kinematics from images alone. For our experiments we render the original RPL datasets and consider two environments from the D4RL benchmark [24]. The "mixed" task requires operating the microwave, kettle, light switch and slide cabinet and has a small set of successful demos in the offline dataset. The "partial" task, which requires manipulating the microwave, kettle, bottom burner and light switch does not have any trajectories that successfully complete all four objects, but has demonstrations for several configurations which complete up to three objects. We will release this dataset with our project to facilitate the development and testing of vision-based offline RL algorithms.

For the model-free methods, since we use a feedforward network for encoding images, we use a framestack of 3 for all of our model-free experiments. At each timestep $t$, the agent was provided with a history of the previous 3 images (from the offline trajectories during offline training, or from the environment during online training). For COMBO and LOMPO, since the latent dynamics model has a recurrent component and therefore can implicitly retain a history of observations, we did not use any framestacking with the image observations from the environments.

One the Franka Kitchen Environment, we did not use an action repeat, and on the Metaworld environments and data we used an action repeat of 2. For the online finetuning experiments, we used the following procedure: roll out the current policy in the environment for a single episode, add that episode to the replay buffer, and then finetuning the model, critic network, and the policy network. On the Franka Kitchen environment, after each episode we performed 50 gradient steps on each component of each method (eg: model, critic network, and the policy network). For the Metaworld environments, we performed 20 gradient steps after each episode. In total, on the Franka Kitchen environments, we performed $10,000$ gradient steps of offline training and $66,300$gradient steps of

| Environment | Avg. Return | Success Rate |
|---|---|---|
| assembly-v2 | 36.000 | 1.000 |
| bin-picking-v2 | 20.900 | 1.000 |
| box-close-v2 | 25.300 | 1.000 |
| coffee-push-v2 | 36.200 | 1.000 |
| disassemble-v2 | 31.556 | 1.000 |
| door-open-v2 | 15.200 | 1.000 |
| drawer-open-v2 | 48.000 | 1.000 |
| hammer-v2 | 63.333 | 1.000 |
| plate-slide-v2 | 71.100 | 1.000 |
| window-open-v2 | 60.500 | 1.000 |

Table 1: Undiscounted episode returns and success rates in the MetaWorld datasets.

online finetuning. On the Metaworld environments, we performed $1,000$ gradient steps of offline training and $20,000$ gradient steps of online finetuning.

## C.2 Datasets

**Kitchen**

- Number of trajectories: $563$
- Number of transitions: $128,569$
- Average undiscounted episode return: $261.12$
- Average number of objects manipulated per episode: $3.98$

**MetaWorld** All of the MetaWorld datasets have $9 - 10$ trajectories and $1,010$ total transitions. The average undiscounted episode returns and success rates are shown in Table 1:

## C.3 Model Based Methods

MOTO uses the model architecture from [32]. For the convolutional image encoder network, we use the following hyperparameters:

- channels: $(48, 96, 192, 384)$
- kernel sizes: $(4, 4, 4, 4)$
- strides: $(2, 2, 2, 2)$
- padding: `VALID`
- four final MLP layers of size: $400$

The decoder network consists of Deconvolution/Transpose convolution layers with the following hyperparameters:

- four initial MLP layers of size: $400$
- channels: $(128, 64, 32, 3)$
- kernel sizes: $(5, 5, 6, 6)$
- strides: $(2, 2, 2, 2)$
- padding: `VALID`

MOTO was trained using a model learning rate of $1 \times 10^{-4}$. The critic and policy network learning rates are $8 \times 10^{-5}$. The batch size for model training is 16 and the batch size for agent training is 128. We also used a filtered behavioral cloning factor of 10 and a disagreement penalty factor of 10.

The latent dynamics model is represented using an RSSM [58] with an ensemble size of 7 models. All other hyperparameters are the default values in the DreamerV2 repository.

The DreamerV2 baseline uses the same hyperparameters as used for MOTO (excluding the behavioral cloning factor and the disagreement penalty factor).

COMBO [12] and LOMPO [19] were run using the image-based implementations from the LOMPO repository. For the image encoder network of the model, we use the default convolutional encoder architecture, which has the following hyperparameters:

- channels: $(32, 64, 128, 256)$
- kernel sizes: $(4, 4, 4, 4)$
- strides: $(2, 2, 2, 2)$
- padding: `VALID`
- final MLP layer size: $1024$

Similarly, the decoder network consists of Deconvolution/Transpose convolution layers with the following hyperparameters:

- initial MLP layer size: $1024$
- channels: $(128, 64, 32, 3)$
- kernel sizes: $(5, 5, 6, 6)$
- strides: $(2, 2, 2, 2)$
- padding: `VALID`

The latent dynamics model is represented using an RSSM [58] with an ensemble size of 7 models.

Both COMBO and LOMPO were trained using a model learning rate of $6 \times 10^{-4}$, and critic network learning rate of $3 \times 10^{-4}$, and a policy network learning rate of $3 \times 10^{-4}$. The batch size for model training is $64$ and the batch size for agent training is $256$. For COMBO, we use a conservatism penalty factor of $\alpha = 2.5$, and for LOMPO we use a disagreement penalty factor of $\lambda = 5$.

### C.4    Model Free Methods

The model free baselines (IQL [3], CQL [4], SAC [49], BC) were run using the JAXRL2 framework [59]. For all policy networks, critic networks, and value networks, we used a feed-forward convolutional encoder network architecture from the D4PG method [60], with the following hyperparameters:

- channels: $(32, 64, 128, 256)$
- kernel sizes: $(3, 3, 3, 3)$
- strides: $(2, 2, 2, 2)$
- padding: `VALID`
- final MLP layer size: $50$

This encoder was then followed by two MLP layers of size $256$, followed by a final output layer of size 1 (for critic and value networks) or of size `action-dim` for policy networks. ReLU activations were used between each layer.

We use a discount factor $\gamma = 0.99$ and a batch size of $256$ for all of the methods, as well as a learning rate of $3 \times 10^{-4}$ for all policy, critic, and value networks. We also used a soft target update

for critic and value networks with a factor of $\tau = 0.005$. For CQL we set the conservatism penalty factor $\alpha = 5$, and for IQL we set the expectile hyperparameter $\tau = 0.5$ and the inverse temperature hyperparameter $\beta = 3$, which are the default values in JAXRL2. For all other hyperparameters, we used the default values in JAXRL2.

