# OpenReview forum: "MOTO: Offline Pre-training to Online Fine-tuning for Model-based Robot Learning"
_robot-learning.org/CoRL/2023/Conference — CoRL 2023 Poster_

### Official Review · Reviewer_ZAHx · 2023-07-17

**Confidence:** 4
**Originality:** Good
**Technical Quality:** Very Good
**Clarity Of Presentation:** Very Good
**Impact:** 3

**Recommendation:**

Weak Accept: I recommend accepting the paper, but will not argue for my recommendation if the majority of other reviewers have a different opinion.

**Review:**

**Quality**: The quality of this work is good. The paper is well-structured, the methodology is clearly explained, and the results are backed by comprehensive experiments. The paper combines several existing methods to build a working framework with offline pretraining and online fine-tuning setting, where the setting itself is very important.

**Clarity**: The paper is well-written and the clarity is good.

**Originality**: The work is original in its approach to merging offline pre-training and online fine-tuning in a model-based reinforcement learning context. The MOTO algorithm introduces innovative techniques like model-based value expansion, epistemic uncertainty corrections, and policy regularization.

**Significance**: This work is good as it pushes the boundaries of reinforcement learning, particularly in the realm of robot tasks using high-dimensional observations. The MOTO algorithm's superior performance in comparison to existing methods demonstrates its potential impact in this field. However no real-robot experiment is conducted. The paper follows many existing methods from offline RL, model-based RL and imitation learning and well combines them under the offline pretraining and online fine-tuning setting.

**Strengths**:
1. **High Performance**: MOTO achieves better results than existing methods on several benchmark tasks, showcasing its effectiveness.
2. **Efficient Data Use**: MOTO efficiently uses offline data for pre-training, making it more data-efficient than many existing methods.
3. **Versatility**: The algorithm shows versatility by performing well on a variety of tasks.


**Weaknesses**:
1. **Limitations in Generalization**: The algorithm might struggle with generalization outside of the available data distribution. This limitation could affect its performance on tasks with unexpected or unseen scenarios.
2. **Sparse Reward Tasks**: The approach may be difficult to adapt to tasks involving more complex or non-sparse rewards.
3. **Transferability Concerns**: It is uncertain whether the scheme used to manage model-based epistemic uncertainty can transfer to more complex architectures, such as Transformers.
4. **Lack of Real-World Testing**: The algorithm is tested on benchmarks and not in real-world settings, so its practical applicability and robustness are yet to be determined.

**Quality Of The Limitations Section:**

Limitations are addressed clearly

**Questions For Rebuttal:**

* The figure 1 is in extremely poor quality.
* In line 113 page 4, these is a missing reference.
* In line 144, when discussing "using only uncertainty-penalized reward" you can refer back to Eq. 2 for clarity.
* The authors should clarify how the size and quality of the offline dataset affect the performance of MOTO. This would provide more insights into the data efficiency of the method.
* We can also discuss the "ratio" of offline pretraining and online fine-tuning. A question is when increasing the pretraining dataset while decreasing the online interactions, could the model achieve similar result? If not then what is the relationship between the size of two dataset?

**Robotics Focus:**

Highly relevant to robotics but no hardware experiments

**Summary Of Paper:**

The paper introduces MOTO (Model-based Offline-To-Online) as a new fine-tuning approach for online reinforcement learning in the context of robot tasks using high-dimensional observations.

The authors argue that existing high-dimensional, model-based offline reinforcement learning methods aren't suitable for offline-to-online fine-tuning due to issues like distribution shifts, off-dynamics data, and non-stationary rewards.

MOTO is proposed as an on-policy model-based method that can efficiently reuse prior knowledge collected offline, while also being able to adapt online to novel tasks. MOTO has three parts: model-based value expansion, uncertainty-aware latent dynamic model and behavior-regularized policy optimization (adding a weighted imitation learning terms).

The paper demonstrates that MOTO outperforms recent model-free and model-based methods on several benchmark tasks, helping to improve both performance and sample efficiency.

**Summary Of Recommendation:**

This work connects established method from offline RL, model-based RL and imitation learning and showcases a working framework toward offline pretraining and online fine-tuning. I recommend acceptance as how to utilize large-scale dataset to facilitate real-world robotics is a very important problem and this work provides a method.

---

### Official Review · Reviewer_Ar1n · 2023-07-18

**Confidence:** 3
**Originality:** Good
**Technical Quality:** Good
**Clarity Of Presentation:** Good
**Impact:** 4

**Recommendation:**

Weak Accept: I recommend accepting the paper, but will not argue for my recommendation if the majority of other reviewers have a different opinion.

**Review:**

Strengths:
- The problem of the adaptation of model-based RL algorithms is quite an important one, and going from the offline to the online setting has been relatively understudied. As such, this paper deals with a crucial issue in RL and could potentially have a high impact.
- The method proposed by the authors shows strong empirical results, especially in terms of generalization capabilities on Franka Kitchen.
- Related work is nicely described.

Weaknesses:
- The value expansion seems to be the central contribution proposed in the paper, but its motivation is not sufficiently discussed and its impact is not properly examined. Why is it that value expansion would help with the replay problem? How is your approach different from the value expansion in Dreamer-v2 you base on (they also use GAE-like expansion)? How is the horizon length set and how much does it impact the results? I think the paper would benefit from a more detailed discussion and more ablation studies on this part.
- The impact of the architecture on the results has not been studied. I find this important, since MOTO uses Dreamer-like architecture, while all the baselines use a smaller architecture with a higher-dimensional latent space. Currently, it is not clear whether the improvements offered by Dreamer and MOTO are solely due to the algorithmic changes or also due to a better-suited architecture.
- The paper does not discuss the computational expenses of the method. Given the complexity of the method, it would be good to have an insight into how expensive it is to train it, e.g. how many GPU hours it took to train it vs. the rest of the baselines.
- There are no real-world experiments in the paper.
- Minor point: the presentation of the paper should be improved in some regards:
   - Figure 4 mentions  "No BC", but this abbreviation is not explained in the text. I assume you mean Behavioral Cloning, i.e. behaviour prior policy regularization?
   - It's difficult to tell apart colors in Figure 4 (dark blue vs black).
   - Line 58: "kicthen domain" -> kitchen
   - Line 76: "is sampled form" -> "sampled from"
   - Line 111: "where α is a tarde-off" -> "trade-off"

**Quality Of The Limitations Section:**

Limitations are addressed clearly

**Questions For Rebuttal:**

See the weaknesses section above. In particular, I would appreciate:
- A more thorough discussion of the value expansion technique.
- An investigation of the impact of the architecture
- A discussion of the computational costs of the method.
- Fixing the minor presentation errors.

**Robotics Focus:**

Highly relevant to robotics but no hardware experiments

**Summary Of Paper:**

The authors consider the problem of online fine-tuning model-based RL approaches that were pre-trained on offline data. They argue that existing methods do not solve this problem sufficiently, as offline model-based RL algorithms fail at fine-tuning. The proposed solution relies on introducing model-based value expansion as a way to train the policy and the value networks and properly regularizing the policy during fine-tuning. The authors show that using this approach they can outperform prior work on the Meta-World suite as well as the challenging Franka Kitchen tasks. Finally, the authors present ablation studies and discuss the limitations of their study.

**Summary Of Recommendation:**

The quality of the paper is quite high, but there are some important issues that I think should be fixed before publishing the paper. As such, for now, I decided to go with weak reject. I would be happy to increase the score after addressing these issues.

---

### Official Review · Reviewer_KHbi · 2023-07-18

**Confidence:** 3
**Originality:** Fair
**Technical Quality:** Good
**Clarity Of Presentation:** Very Good
**Impact:** 3

**Recommendation:**

Weak Reject: I recommend rejecting the paper, but will not argue for my recommendation if the majority of other reviewers have a different opinion.

**Review:**

Strengths:
- The paper is very easy to follow and well-written.
- The proposed method is clean and each design choice is well-motivated
- The proposed method achieves good performance on a set of tasks compared to baselines, especially on the Franka Kitchen task from images.

Weaknesses:
- Although the proposed method works well in the tested tasks, each proposed component is not new and has already used in previous methods, including
    - The Q function is jointly trained by both model-based loss and model-free loss. This idea is similar to the loss used in, e.g., [1]
    - Considering uncertainty when unrolling the trained dynamics model. This idea is similar to [2] and several other model-based offline RL papers.
    - The idea of adding behavior cloning when updating the policy is also used before, e.g., [3][4][5]

All these methods are known to work in their settings and one would expect the combination of these techniques can work in the offline-to-online setting. I can understand that combining these techniques and demonstrating promising results on a set of tasks take effort, however, the insights obtained from this paper are limited.


[1] Ma, Xiao, et al. "Contrastive variational reinforcement learning for complex observations." Conference on Robot Learning. PMLR, 2021.
[2] Rafailov, Rafael, et al. "Offline reinforcement learning from images with latent space models." Learning for Dynamics and Control. PMLR, 2021.
[3] Fujimoto, Scott, and Shixiang Shane Gu. "A minimalist approach to offline reinforcement learning." Advances in neural information processing systems 34 (2021): 20132-20145.
[4] Zhao, Yi, et al. "Adaptive behavior cloning regularization for stable offline-to-online reinforcement learning." arXiv preprint arXiv:2210.13846 (2022).
[5] Siegel, Noah Y., et al. "Keep doing what worked: Behavioral modelling priors for offline reinforcement learning." arXiv preprint arXiv:2002.08396 (2020).

**Quality Of The Limitations Section:**

Limitations are addressed clearly

**Questions For Rebuttal:**

Please check the weaknesses in the Review section.

**Robotics Focus:**

Highly relevant to robotics but no hardware experiments

**Summary Of Paper:**

The paper proposes model-based reinforcement learning in the offline pre-training to online fine-tuning setting. During training, it mainly introduces three techniques to achieve good performance: 1. training the Q function on both dataset and unrolled trajectories by the learned dynamics model; 2. add reward penalty by considering the model uncertainty when unrolling the dynamics model; 3. add behavior prior policy regularization when training the policy. The proposed method achieves good performance on pixel-based offline-to-online tasks.

**Summary Of Recommendation:**

The paper is easy to follow and well-written and each design choice is well-motivated.  The proposed method achieve strong performance on a set of pixel-based tasks. My main concern is that three newly introduced components compared to Dreamer V2 are already used in previous works, especially two of them are known to work in offline reinforcement learning settings, e.g., considering model uncertainty and adding behavior cloning loss. People would expect combining these techniques works in the offline-to-online setting, which limits the significance of this work. Therefore, my recommendation is Weak Reject.

---

### Official Review · Reviewer_kYGq · 2023-07-20

**Confidence:** 3
**Originality:** Good
**Technical Quality:** Very Good
**Clarity Of Presentation:** Very Good
**Impact:** 3

**Recommendation:**

Weak Accept: I recommend accepting the paper, but will not argue for my recommendation if the majority of other reviewers have a different opinion.

**Review:**

Paper is well written and states its contributions clearly. The proposed algorithm is a novel combination of existing components, which are well suited to solve the tasks selected for the experiments.

Impressively small dataset of offline trajectories for training in the MetaWorld task.

Additionally, Impressive results on solving the Franka Kitchen environment from pixels.


While solving robotic manipulation environments from pixels is a step towards usability of model-based RL on real-world robots, it seems unlikely that the proposed algorithm will be used in a real-world scenario anytime soon, for a few reasons:

- Task completion reward from the simulator may not be readily available in real-world scenarios
- Real-world visual observations may be much noisier and more diverse than simulated images.
- The reported 500K environment steps required for training are prohibitive for most applications involving real-world hardware.
- Another detriment to real-world robot tasks is mentioned in the limitations - MOTO may deteriorate if the offline dataset is lower-quality or incomplete, which is a common trait of real-world datasets.


Some tiny nit-picks:
- Line 111: tarde → trade
- Broken citation on line 113

**Quality Of The Limitations Section:**

Limitations are addressed clearly

**Questions For Rebuttal:**

In the MetaWorld results, seems like most of the advantage of MOTO comes from its DreamerV2 architecture. Have the authors attempted to insert the DreamerV2 architecture into any of the other model-based offline-RL baselines for comparison?

**Robotics Focus:**

Relevant but unlikely to deploy to hardware in near future

**Summary Of Paper:**

This paper proposes MOTO, a new model-based algorithm for the offline pre-training to online fine-tuning regime. MOTO operates on pixel inputs using the DreamerV2 architecture, with adjustments made to the model-based actor-critic algorithm, such as MVE for the actor and critic losses, and uncertainty regularization based on an ensemble of dynamics models. The proposed method shows favorable results compared to baselines on simulated robotic manipulation tasks.

**Summary Of Recommendation:**

While the approach presented in this paper may not be easily transferrable to real robots, its impressive results on pixel-based robotic manipulation tasks in simulation, as well as its improvement of the state-of-the-art in model-based RL render it a worthy contribution to the robot learning community.

---

### Author Response · Authors · 2023-08-16
**Thank you!**

We would like to thank the reviewers for the detailed and valuable feedback and engaging in a fruitful discussion. We are working to address their suggestions and will include additional results as they become available.

---

### Decision · Program_Chairs · 2023-08-30

**Decision:**

Accept (Poster)

**Comment:**

The paper proposes a model-based method for going from offline to online data in reinforcement learning. Model learning is performed on policy. In experimental evaluation in simulation tasks MetaWorld and Franka Panda Kitchen, the proposed approach is competitive w.r.t. comparison methods. The paper is well written.